# Is the TEX11-.652del237bp Exonic In-Frame Deletion Variant Associated with Azoospermia? The Results of an *In Vitro* and *In Silico* Study

**DOI:** 10.3390/genes16111270

**Published:** 2025-10-28

**Authors:** Morgane Le Beulze, Dorothée Poidatz, Marie Francisco, François-Xavier Madec, Pierre-Henri Benetti, Gabriel Livera, François Vialard

**Affiliations:** 1Equipe RHuMA UMR 1198, BREED, UFR Simone Veil—Santé, UVSQ, 78180 Montigny le Bretonneux, France; dorothee.poidatzbenetti@gmail.com (D.P.); marie.francisco@agroparistech.fr (M.F.); 2Stabilité Génétique Cellules Souches et Radiations, Université Paris Cité, CEA, LDG/IRCM/IBFJ, 92260 Fontenay-aux-Roses, France; gabriel.livera@cea.fr; 3Stabilité Génétique Cellules Souches et Radiations, Université Paris-Saclay, CEA, 92260 Fontenay-aux-Roses, France; 4Department of Urology, Foch Hospital, 92150 Suresnes, France; f.madec@hopital-foch.com; 5UFR Microbiologie et Génétique Moléculaire, AgroParisTech, 91123 Palaiseau, France; benetti@agroparistech.fr; 6UFR Simone Veil Santé, UVSQ, 78180 Montigny-le-Bretonneux, France

**Keywords:** TEX11, azoospermia, maturation arrest, *in vitro* model, deletion

## Abstract

**Background:** In 2015, it was discovered that mutations in the *TEX11* gene are associated with azoospermia in general and meiotic maturation arrest in particular. TEX11 is a component of the ZZS complex (comprising Zip2-, Zip4- and Spo16 and originally described in *Saccharomyces cerevisiae*). During meiosis, this complex is required for the promotion of double-strand break (DSB) repair and thus the maintenance of genomic integrity. Since the initial discovery, several variants and deletions in *TEX11* have been reported in patients with spermatogenesis defects. However, many of these new variants have not been functionally validated, which makes it difficult to confirm their direct impact on meiosis. The exonic in-frame deletion *TEX11-*c.652del237bp has been recurrently identified in infertile men. However, mice models carrying this deletion remain fertile—suggesting that these models may not faithfully replicate human meiotic phenotypes. To address this discrepancy, we functionally validated the *TEX11*-c.652del237bp variant *in vitro*. **Methods**: After amplification in *Escherichia Coli* DH5α, the pIRES2-EGFP plasmid containing either the wild-type *TEX11* sequence or the *TEX11-*c.652del237bp sequence was transfected into the HEK293 human embryonic kidney cell line. qPCR and Western blot analyses were then used to evaluate the presence and expression levels of *TEX11* mRNA and TEX11 protein. **Results**: The qPCR and Western blot analyses showed that truncated mRNA and protein were produced in cells transfected with the c.652del237bp variant. Hence, the deletion probably leads to the transcription and translation of *TEX11* in human testis. Furthermore, *in silico* modeling suggested that the deletion does not have a significant impact on the ZZS complex. **Conclusions**: Our *in vitro* and *in silico* data demonstrate that the c.652del237bp in-frame deletion results in a truncated TEX11 protein and thus question the deletion’s pathogenic role in human meiosis. However, the absence of a meiotic phenotype in the corresponding mouse model is suggestive of species-specific differences in TEX11 endogenous function. Further studies (such as co-immunoprecipitation experiments with other ZZS complex proteins) are needed to fully assess the functional impact of *TEX11-*c.652del237bp. These experiments might also provide novel insights into the specific role of the TEX11 SPO22 domain in human spermatogenesis.

## 1. Introduction

The World Health Organization considers infertility (defined as the inability to conceive after 12 months or more of regular, unprotected sexual intercourse) to be a major public health concern. Infertility affects approximately 15% of couples of childbearing age and is of male origin in about half of these cases [1]. The main causes of male infertility are related to low sperm counts and poor sperm quality. Low sperm counts are associated with an elevated frequency of chromosomal abnormalities. Karyotyping and Y chromosome microdeletion screening [2,3,4] are the primary first-line genetic screening techniques for male infertility and can be used to diagnosis Klinefelter syndrome (a 47,XXY karyotype) [5], chromosomal rearrangements [2], and Y chromosome microdeletions. At present, the only contraindications to testicular sperm extraction (TESE) are AZFa and/or AZFb microdeletions (leading, respectively, to Sertoli cell-only syndrome and spermatogenic maturation arrest [6]) and a 46,XX karyotype in a phenotypic male [7].

Given that no more than 20% of men with nonobstructive azoospermia (NOA) have chromosomal abnormalities, it has been hypothesized that other spermatogenesis related gene mutations are located elsewhere in the genome. This hypothesis has been supported by (i) the diagnosis of azoospermia in inbred families and (ii) observations of the many murine models of azoospermia described in the literature. Whole-genome analyses (especially array comparative genomic hybridization (aCGH), whole-exome sequencing (WES), and whole-genome sequencing (WGS)) have revealed associations between gene defects and spermatogenesis failure or NOA [8,9,10,11]. Although defects in Testis-expressed gene 11 (*TEX11*) are the most frequently identified variants in patients with NOA and spermatogenesis maturation arrest at the pachytene stage, the list of gene mutations leading to NOA is lengthening. A large number of genes are likely to be involved in spermatogenesis [12], including the TEX11 partners of ZZS (an acronym for yeast proteins Zip2-Zip4-Spo16): SHOC1 (Shortage In Chiasmata 1), M1AP (Meiosis 1 Associated Protein), and SPO16). At present, the gene variants most frequently reported in literature are *TEX11*-c.652del237bp (originally identified in three patients with NOA) and *M1AP*-c.676dup [13].

Although whole-genome analytical techniques were initially used to help diagnose the etiology of male infertility, they can also be applied as prognostic tools for evaluating the likelihood of a positive TESE. In general, we need to find ways of preventing unnecessary TESEs in NOA patients with genetic abnormalities like AZFa and/or b microdeletions or a 46,XX karyotype. Furthermore, we need to obtain a clear, functional validation of the identified gene defects, in order to provide accurate genetic counselling and estimate the likelihood of a positive TESE. Various animal models and *in vitro* models have been used for functional validation. For example, *Trypanosoma brucei* has been used to validate genetic variants identified in the multiple morphologically abnormal flagellum (MMAF) of spermatozoa [14] because the latter is very similar to the trypanosome’s flagellum. Likewise, various murine models have been used to validate human genetic defects in spermiogenesis (e.g., globozopermia and MMAF). If, however, the animal model for a infertile human variant is fertile and/or does not mirror the human phenotype in other ways, no conclusions can be drawn. It was recently reported that even though *M1AP* loss of function variants (and especially *M1AP*-c.676dup) are associated with lower chromosome recombination rates and poor spermatogenesis, fatherhood is possible through medically assisted reproduction [15]. Hence, diagnosis of these variants is not a counterindication to TESE. However, a number of questions concerning *TEX11*-c.652del237bp remain, and no conclusions can be drawn.

As mentioned above, TEX11 is part of the ZZS complex required for meiotic recombination, a process in which homologous chromosomes exchange genetic material during meiosis. The ZZS complex interacts with proteins from the synaptonemal complex, a protein structure that forms between homologous chromosomes and helps to ensure correct chromosome pairing and crossover formation.

With a view to functionally validating the *TEX11*-c.652del237bp recurrent, exonic, in frame deletion in exons 9 to 12 (TEX11-p.Thr203_Lys281del), we previously generated a mouse model (Tex11Ex9-11del/Y) [16]. Unexpectedly, mutant male mice were fertile. The sperm count, sperm motility, and sperm morphology were all normal. Hence, the equivalent of the human *TEX11*-c.652del237bp variant had no obvious effect on spermatogenesis or fertility in the mouse. There are several possible explanations for the discrepancy between the observations in the mouse and those in human. Firstly, the truncated TEX11 protein (lacking 79 amino acids within the SPO22 domain) might retain sufficient function in mice. Secondly, the SPO22 domain might not be essential for TEX11’s function in murine spermatogenesis. Thirdly, this exonic deletion might not be the cause of azoospermia in the human. Fourthly, the human infertility phenotype might result from the variant’s effects on TEX11 expression. Lastly, structural or conformational differences between murine and human proteins (influenced by the genetic background) might modify the deletion’s functional impact. Taken as a whole, our findings highlighted the complexity of genetic regulation in fertility and underscored the need for further *in vitro* and/or *in silico* clarification of the role of the *TEX11*-c.652del237bp deletion in human spermatogenesis.

To address this question, we evaluated *TEX11* mRNA and protein expression levels *in vitro* after transfecting HEK293 cells with a pIRES2-EGFP plasmid containing either the wild type *TEX11* sequence (*TEX11*-WT) or the *TEX11*-c.652del237bp variant. Furthermore, we modeled the ZZS complex with the TEX11-p.Thr203_Lys281del variant *in silico*.

## 2. Materials and Methods

### 2.1. Plasmid Construction

Constructs were confirmed by DNA sequencing. The *TEX11*-WT and *TEX11-*c.652del237bp DNA plasmids were produced by ProteoGenix (67300 Schiltigheim, France) (Figure 1). Briefly, each genomic sequence was inserted into a pIRES2-EGFP backbone with a cytomegalovirus promoter and the antibiotic resistance genes neomycin and kanamycin. A c-Myc-tag-encoding sequence (GAGCAGAAACTCATCTCAGAAGAGGATCTG) was added to the C-terminal region of each *TEX11* sequence. The *TEX11*-WT and *TEX11*-c.652del237bp plasmids comprised 8102 and 7865 bp, respectively.

### 2.2. Plasmid Amplification and Purification

Using a heat shock protocol, competent *E. coli* DH5α bacteria were transformed with the *TEX11*-WT and *TEX11*-c.652del237bp plasmids. Bacteria having incorporated the plasmid were selected by overnight culture in a Petri dish containing LB medium supplemented with kanamycin. Colonies were then collected for amplification in liquid LB medium with neomycin or kanamycin, as appropriate. Plasmids were purified with a Miniprep Nucleospin^®^ Plasmid kit (Macherey-Nagel, 52355 Dueren, Germany) according to the manufacturer’s instructions. The cleanness of samples and their concentration were measured using the plate reader Infinite^®^ M200 (TECAN, 8708 Männedorf, Switzerland).

### 2.3. Cell Culture and Plasmid Transfection

We decided to use the HEK293 human embryonic kidney cell line because it (i) is the cell line most frequently used in scientific experiments worldwide, (ii) expresses few testis specific genes and (iii) does not express *TEX11* (according to the Human Protein Atlas (https://www.proteinatlas.org/, accessed on 9 October 2025).

HEK-293 (ATCC) cells were cultured in a 5% CO_2_ humidified atmosphere at 37 °C. Dulbecco’s modified Eagle’s medium high glucose pyruvate (Gibco, Thermo Fisher Scientific, 02451 Waltham, MA, USA) was supplemented with 10% fetal bovine serum (Gibco, Thermo Fisher Scientific, 02451 Waltham, MA, USA) and 1% penicillin-streptomycin (Gibco). Plasmids were transfected with Lipofectamine 2000 (Invitrogen, Thermo Fisher Scientific, 02451 Waltham, MA, USA) in 96-well plates, according to the manufacturer’s instructions. The transfection efficiency was quantified using an Infinite^®^ M200 (TECAN, 8708 Männedorf, Switzerland) plate reader at a wavelength of 488 nm (corresponding to the emission spectrum of green fluorescent protein).

The conditions selected for transfection were as follows. Two days before transfection, 20,000 HEK293 cells per condition were grown in media supplemented with 10% FBS and 1% antibiotics, as described above. On the day before transfection, the cells were grown in medium supplemented with 10% FBS but not with antibiotics. On the day of the transfection, the cells were grown in medium supplemented with 2% FBS but not with antibiotics. The cells were transfected with 0.5 µL lipofectamine and 0.1 µg of plasmid in 100 µL of cell transfection medium for 4 h. The transfection was terminated by switching to medium supplemented with 10% FBS and 1% antibiotic. The cells were cultured for a further 48 h period before analysis.

### 2.4. RT-qPCR

The RT-PCR procedure has been described previously [17]. Briefly, total RNA was isolated after cell confluence, extracted with Trizol solution, purified on Qiagen columns (RNeasy mini-kit, Qiagen, 40721 Hilden, Germany), and treated with 1 μL of DNase I. The purified RNA was quantified and checked for quality and intactness. Each sample was reverse transcribed twice, and the products were pooled. The cDNA was then diluted in DNase free water and stored at 80 °C until required. Specific primers upstream of the deleted region and spanning the deleted region (Table 1) were used to check the presence of the RNA and the deletion, respectively. All PCRs were carried out in duplicate, under the following cycling conditions: denaturation at 95 °C for 10 min, 45 denaturation cycles at 95 °C for 15 s, annealing, and DNA synthesis at 60 °C for 15 s. The genes coding for ribosomal protein L13A (RPL13A), TATA binding protein (TBP), and beta-2 microglobulin were chosen as references. For each sample, the concentration ratio (target RNA/three reference mRNAs) was calculated using CFX Manager software version 2.5 (BioRad, Hercules, CA 94547, USA) and expressed in arbitrary units.

### 2.5. Western Blots

The Western blotting procedure has described previously [18]. Briefly, the cells were washed, and proteins were extracted from cell samples in a Tris (20 mM)-EDTA (0.2 mM)-NaCl (137 mM) buffer containing nonidet P4 (1%), glycerol (10%), and various proteinase and phosphatase inhibitors (aprotinin (5 µg/mL), 4-(2-aminoethyl)benzenesulfonyl fluoride hydrochloride (0.1 mg/mL), leupeptin (12.5 µg/mL), sodium orthovanadate (1 mM), beta-glycerophosphate (30 mM), and sodium fluoride (10 mM). After centrifugation, the protein content was assayed using Bradford’s method and bovine serum albumin as the standard. For each sample, 50 μg of protein were deposited on a 4–20% Tris-Glycine Mini gel (Novex ^TM^ Wedgewell ^TM^, ThermoFisher Scientific, 02451 Waltham, MA, USA) for migration in Tris Glycine SDS 1X buffer at a constant intensity of 200 mA per gel. The proteins were transferred to a polyvinylidene difluoride membrane, which was then stored for 2 h at room temperature in Tris HCl (50 mM)/NaCl (2.5 mM) supplemented with dried milk (5%) and Tween (0.05%) (referred to as TBS Tween). The membranes were then incubated overnight at 4 °C in Tris-buffered Tween saline (Tris HCl (50 mM), NaCl (15 mM), Tween 20^®^ (5%)) containing primary rabbit polyclonal antibodies (diluted 1:1000; Novus bio NBP1-80692) and then incubated with anti-rabbit IgG conjugated to horseradish peroxidase (diluted 1:10,000; Interchim UP559721, Montluçon, France). The blot was revealed with a Pierce ECL 23 Western blotting Substrate kit (Thermo Fisher Scientific, 02451 Waltham, MA USA), and the chemiluminescence signal was quantified with a ChemiDoc reader (Biorad, Hercules, CA 94547, USA).

### 2.6. In Silico Modeling

We reasoned that *in silico* structural modeling of the TEX11-p.Thr203_Lys281del variant could complement our *in vitro* findings and provide additional insights. Computational approaches allow the prediction of (i) protein conformational changes induced by exonic deletions and (ii) the changes’ potential impact on protein stability, interaction domains, and partner binding. Since TEX11 exerts its role in meiosis within the ZZS complex (together with M1AP, SPO16, and SHOC1), structural modeling might indicate whether the loss of 79 amino acids in the SPO22 domain alters TEX11’s folding, induces steric hindrance, and/or disrupts protein–protein interaction surfaces. This type of analysis can help determine whether a truncated protein retains a degree of function or, in contrast, leads to defective assembly of the recombination machinery—possibly explaining the discrepancy between the human azoospermia phenotype and the fertility observed in the mouse model.

The structures of human TEX11 and its partners were predicted with AlphaFold2 [19] and ChimeraX 1.9 software. On the AlphaFold error plot generated by ChimeraX, the residues are shown on the *x*-axis and *y*-axis. In AlphaFold, each pair of residues is assigned a predicted aligned error (PAE), the value and color of which indicate the level of confidence in the relative position.

## 3. Results

### 3.1. mRNA Levels in Transfected Cells

We used RT-qPCR assays to measure *TEX11* mRNA expression in HEK293 cells (Figure 2). Using primers located upstream of the deleted region, we found that HEK293 cells transfected with a plasmid *TEX11*-WT and those transfected with the *TEX11*-c.652del237bp plasmid expressed *TEX11* to a similar extent. Using primers spanning the deleted region, we confirmed that mRNA from cells transfected with the *TEX11*-*c.652del237bp* plasmid lacked exons 9 to 11. None of the negative controls showed amplification. These results confirmed the presence of the deletion in the *TEX11*-*c.652del237bp* plasmid but indicated that the mutation does not affect the overall mRNA abundance in transfected cells.

### 3.2. Expression of TEX11 Proteins

To evaluate potential RNA decay, we performed Western blots with an anti-TEX11 antibody (Figure 3). In blots of HEK293 cells transfected with the *TEX11*-WT plasmid, we detected a ~100 kDa TEX11 protein; this is consistent with the expected size of 108 kDa. Interestingly, we observed a truncated TEX11 protein (approximately ~85 kDa in size) in HEK293 cells transfected with the *TEX11*-c.652del237bp plasmid. All the controls were negative.

### 3.3. In Silico Modeling

The predicted template modelling score was 0.41 (a value considered to be “medium”) for both the TEX11-WT ZZS complex and the TEX11-p.Thr203_Lys281del ZZS complex.

The region found to be deleted in TEX11-p.Thr203_Lys281del (shown in red in Figure 4a) did not appear to interact with the other ZZS complex proteins. As this domain does not participate in the interaction between TEX11 and its partners, we confirmed the absence of difference in the conformation of the ZZS complex containing the TEX11-p.Thr203_Lys281del protein (Figure 4b).

To further confirm the absence of changes, we analyzed the interaction error plots for TEX11-WT and TEX11-p.Thr203_Lys281del (Figure 4c,d). The human TEX11 protein interacts with SHOC1, M1AP and SPO16 through its C-terminal (Figure 4a). The TEX11 regions with a high PAE for the interaction with SHOC1 (indicated by * in the figure) and M1AP (indicated by ** in the figure) were not modified by the presence of the deletion—suggesting that the ZZS complex is probably functional.

## 4. Discussion

Since the first report in 2015 [11], many TEX11 variants have been described in the literature [20,21]. After chromosomal abnormalities and Y chromosome microdeletions, *TEX11* defects are now considered to be the third most frequent cause of male infertility (in 2–3% of infertile men). In the vast majority of these cases, *TEX11* variants are associated with a maturation arrest in spermatogenesis.

*TEX11-*c.652del237bp (with a 79–amino acid deletion spanning exons 9 to 11: p.Thr203_Lys281del) is the most frequently reported variant in this context. However, the variant is not listed in the gnomAD database (https://gnomad.broadinstitute.org/, version 4.1.0). Initially, the variant was reported in association with mixed testicular atrophy [11], which raised questions about its impact on the testicular phenotype and the likelihood of a positive TESE. To address this issue, we previously generated a mouse model with the same deletion and evaluated the effects on spermatogenesis [16]. Unexpectedly, the mice were fertile, with a normal sperm count and a normal litter size. Both *TEX11* mRNA and TEX11 protein were detectable.

To investigate these unexpected results, we established an *in vitro* model of *TEX11* mRNA and protein expression in human cells (the HEK293 cell line) and compared the results with our findings in mice. The HEK293 cells were transfected with plasmids encoding *TEX11*-WT or the *TEX11*-c.652del237bp variant, and mRNA levels were quantified. Similar expression levels were observed for the two conditions, indicating the absence of mRNA decay for the variant. Western blot analysis confirmed the presence of TEX11-p.Thr203_Lys281del protein (albeit truncated, when expressed by the *TEX11*-c.652del237bp plasmid) under both conditions. The *in silico* modeling results do not suggest that the deletion has a significant impact on the ZZS complex.

Together with our mouse data, these *in vitro* and *in silico* findings raise questions about the pathogenic role of the in frame *TEX11*-c.652del237bp deletion in spermatogenesis because the defect does not abolish TEX11 protein expression. However, given that these experiments were conducted in mice and in the HEK293 cell line, the role of this variant in the human azoospermia phenotype remains subject to debate. The truncated TEX11-p.Thr203_Lys281del protein (lacking 79 amino acids within the SPO22 domain) might be partially functional, and its effects might also be influenced by the genetic background. Compensatory molecular mechanisms might mitigate the functional impairment in mice but not in humans, which could account for species specific differences. Nevertheless, in view of all the available evidence, it is difficult to conclude that the *TEX11*-c.652del237bp variant necessarily precludes a positive TESE. While the identification of the *TEX11*-c.652del237bp variant might explain an infertility phenotype, its presence should not constate a strict contraindication to TESE.

It is also noteworthy that the two patients reported with *TEX11*-c.652del237bp were not assessed with WES or WGS [11,22]. As the diagnosis was based on aCGH, another genetic etiology cannot be ruled out because variants in more than 50 different genes are reportedly associated with azoospermia. Even though the presence of the in frame *TEX11*-c.652del237bp variant might well explain the infertility of the three patients described in the literature, another genetic etiology would have to be ruled out with WES or WGS. Furthermore, aCGH is now an outdated diagnostic technique, given that WES and WGS can identify both copy number variations and single nucleotide changes.

In the absence of a WGS/WES analysis, further experiments are needed to determine the true impact of *TEX11*-c.652del237bp on spermatogenesis. In particular, co-immunoprecipitation studies with the three human TEX11 partners of the ZZS complex (M1AP, SPO16, and SHOC1) would probably be informative. Although *in silico* modeling suggests that the deletion does not have a significant impact on the ZZS complex, a potential negative effect on the protein’s conformation *in vitro* cannot be ruled out. Immunoprecipitation experiments might reveal whether the *TEX11*-c.652del237bp variant alters the overall conformation of the complex and thereby disrupts not only TEX11–partner interactions but also partner–partner interactions. This mechanism might explain the infertility phenotype observed in humans. A similar approach has already been applied to another recurrent variant associated with male infertility (the *M1AP*-c.676dup variant), using the same HEK293 cell model [15].

## 5. Conclusions

The impact of the *TEX11*-c.652del237bp variant (encodes a TEX11-p.Thr203_Lys281del protein lacking 79 amino acids within the SPO22 domain) on spermatogenesis is still subject to debate. Neither the mouse model nor the HEK293 transfection study enables us to draw definitive conclusions; further experiments (e.g., co-immunoprecipitation with all known TEX11 partners) are required. At present and in contrast to cases of 46,XX males or patients with AZFa and/or AZFb microdeletions, there is no strong evidence to contraindicate an initial TESE in patients carrying the *TEX11*-c.652del237bp variant.

## Figures and Tables

**Figure 1 genes-16-01270-f001:**
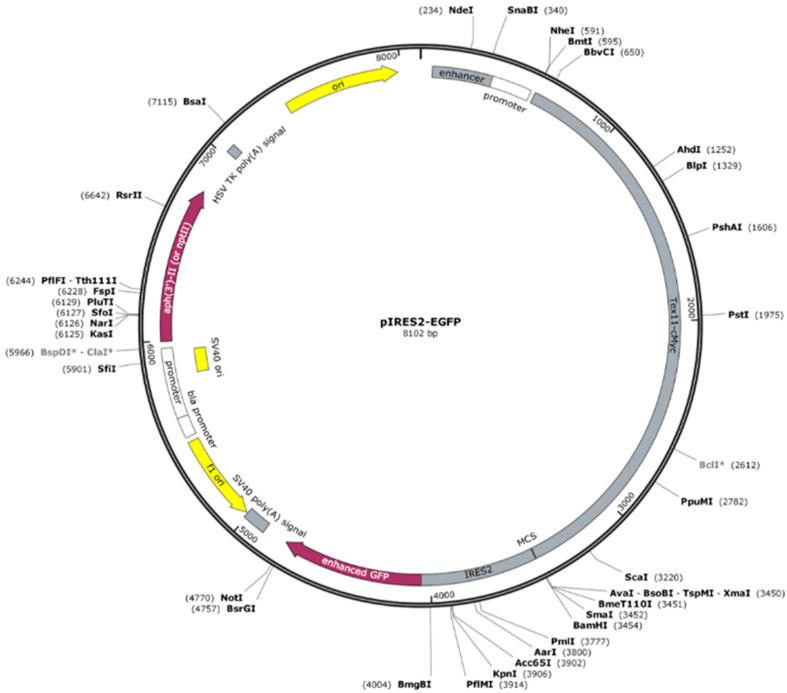
Schematic representation of the pIRES2-EGFP backbone with the *TEX11*-cMYC insertion.

**Figure 2 genes-16-01270-f002:**
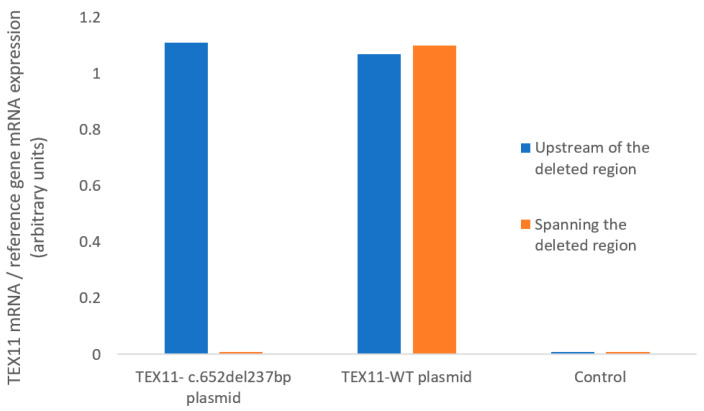
mRNA expression, quantified with primers upstream of the deleted region or spanning the deleted region.

**Figure 3 genes-16-01270-f003:**
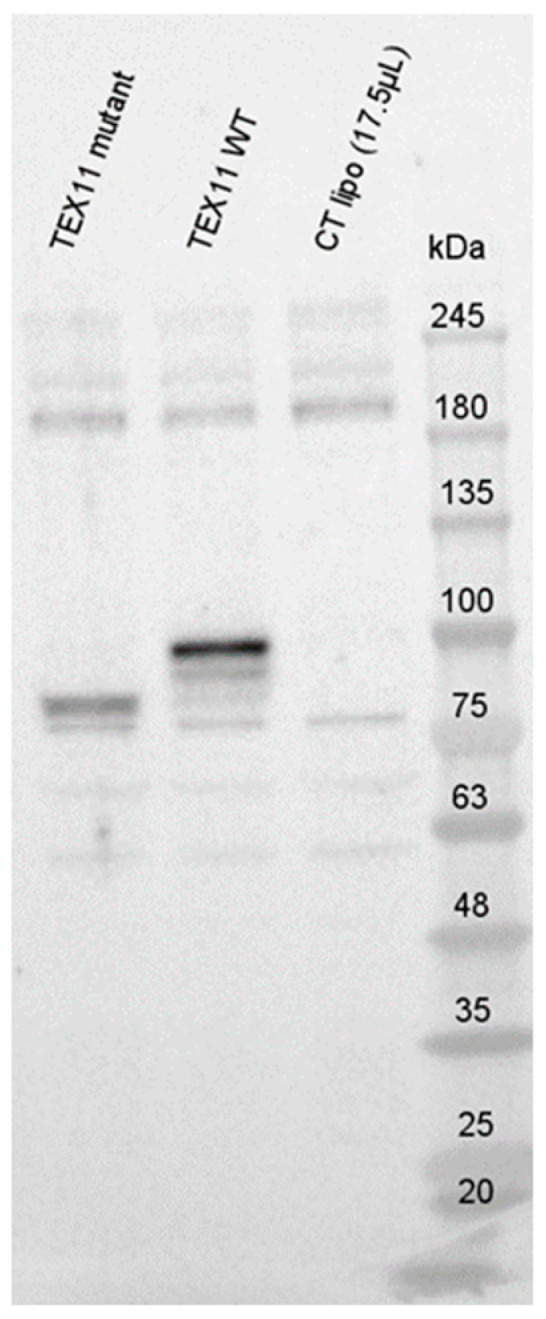
Expression of TEX11 proteins.

**Figure 4 genes-16-01270-f004:**
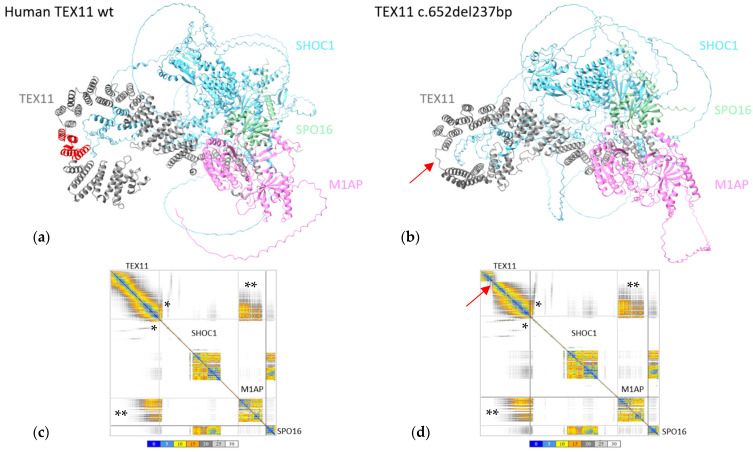
*In silico* protein modeling of human TEX11-WT and TEX11-p.Thr203_Lys281del complexes. (**a**) Modeled structure of the ZZS complex with TEX11-WT complex. The region deleted in TEX11-p.Thr203_Lys281del is shown in red. (**b**) Modeled structure of the ZZS complex with TEX11-p.Thr203_Lys281del. The red arrow indicates the position of the deletion. (**c**) The interaction plot for TEX11-WT. (**d**) The interaction plot for TEX11-p.Thr203_Lys281del. A low PAE (shown in blue on the error plot) indicates that the residues’ relative positions are predicted accurately. A medium PAE (shown in yellow/orange) indicates a degree of uncertainty about the relative positions in domains or regions. A high PAE (shown in gray) indicates high uncertainty about the relative positions in domains or regions. Each square along the diagonal corresponds to a protein in the complex. The rectangles on the edges correspond to the relative position of the residues in a pair of proteins. Within these boxes, the colored areas correspond to the regions of the two proteins that are predicted to interact with each other. The gray shaded areas are potential contact areas, i.e., with low prediction confidence. * The region through which TEX11 and SHOC1 interact. ** The region through which TEX11 and M1AP interact.

**Table 1 genes-16-01270-t001:** Details of the primers.

Primers	Forward	Reverse	PCR Product (bp)
upstream of the deleted region	5′ ACAACTTTGGAGTAGAAACCCAGA 3′	5′ AGGTTTACAGCATTGAGAGCCT 3′	202
spanning the deleted region	5′ TGGAACTGGGCACTTACCAT 3′	5′ GGCAAATGAGGCTTCACACA 3′	111

## Data Availability

The original contributions presented in this study are included in the article. Further inquiries can be directed to Morgane Le Beulze (morgane.lebeulze@cea.fr).

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
