# Peer review of "Is the TEX11-.652del237bp Exonic In-Frame Deletion Variant Associated with Azoospermia? The Results of an In Vitro and In Silico Study"

_genes, 2025, doi:10.3390/genes16111270_

Round 1
Reviewer 1 Report
Comments and Suggestions for Authors
The manuscript “After the in vitro modeling of the exon 9 to 11 deletion of TEX11, is it still associated with azoospemia?” provides information on the association of TEX11 with azoospemia. Also, the authors suggested that its exon 9 to 11 deletion probably leads to azoospermia. First, I think the authors should reconsider the title because I do not understand “after the in vitro modeling”. Does that mean in vitro experiments using transfected-HEK293 cells with the DNA of TEX11? Also, it needs to add a specific explanation of the result (especially results in Fig. 4), because the readers cannot understand the analysis of in silico modeling. Furthermore, the authors need to discuss their results carefully and give the obtained information to readers.
Major points
- First, the authors need to add the reason why they focus on TEX11, but not other components (SHOC and SPO16), as a component of the ZZS complex.
- The authors need to discuss the reasons for using HEK293 cells in this study.
- To improve the validity of this study, it is necessary to explain why different phenotypes are obtained between humans and mice by the deletion c.652del237bp
- The results of this study only show the possibility and have not actually been examined on sperm. Are there any plans to conduct in vivo analysis using KO mice or to test using sperm from azoospermia patients?
Minor points
- There are many “-” in the text. What does this mean (Line 87, 91, …)?
- Fig. 4 is difficult to understand. How about explaining it in parts A, B, C, etc.?
- Fig. 4 should be included in the text, not in a subheading (3. In silico modeling) on page 7.
Author Response
We thank the reviewers and editor for their constructive comments and have answered each query below. The line numbers in the text refer to the “clean” version.
Reviewer 1:
The manuscript “After the in vitro modeling of the exon 9 to 11 deletion of TEX11, is it still associated with azoospemia?” provides information on the association of TEX11 with azoospemia. Also, the authors suggested that its exon 9 to 11 deletion probably leads to azoospermia.
First, I think the authors should reconsider the title because I do not understand “after the in vitro modeling”. Does that mean in vitro experiments using transfected-HEK293 cells with the DNA of TEX11?
We previously studied an in vivo (murine) model of c.652del237bp. Considering that the results were unexpected, we decided to perform an in vitro experiment using HEK293 cells. In line with your suggestion, we have revised the title, which is now “Is the TEX11-.652del237bp exonic in-frame deletion variant associated with azoospermia? The results of an in vitro and in silico study”.
Also, it needs to add a specific explanation of the result (especially results in Fig. 4), because the readers cannot understand the analysis of in silico modeling.
Thank you for this comment. We have now added a short paragraph to the Material and Method section and also explain the results in more depth. Please see our answer to Reviewer 2.
Furthermore, the authors need to discuss their results carefully and give the obtained information to readers.
Please see the answers below.
Major points
- First, the authors need to add the reason why they focus on TEX11, but not other components (SHOC and SPO16), as a component of the ZZS complex.
Thank you for this comment on the importance of evaluating all gene variants leading to azoospermia - and not just those in TEX11 or other components of the ZZS complex. However, we first decided to evaluate the TEX11 variant that is mostly frequently described in the literature, i.e. the c.652del237.
We have added two sentences to the Introduction.
Line 76-78: At present, the gene variants most frequently reported in literature are TEX11-c.652del237 (originally identified in three patients with NOA) and M1AP-c.676dup [13].
Line 95-97: Hence, diagnosis of these variants is not a counterindication to TESE. However, a number of questions concerning TEX11-c.652del237 remain, and no conclusions can be drawn.
- The authors need to discuss the reasons for using HEK293 cells in this study.
We decided to use the HEK293 human embryonic kidney cell line because it (i) is the cell line most frequently used in scientific experiments worldwide, (ii) expresses few testis-specific genes and (iii) does not express TEX11 (according to the Human Protein Atlas (https://www.proteinatlas.org/)). This has now been specified
Lines 150-153: We decided to use the HEK293 human embryonic kidney cell line because it (i) is the cell line most frequently used in scientific experiments worldwide, (ii) expresses few testis specific genes and (iii) does not express TEX11 (according to the Human Protein Atlas (https://www.proteinatlas.org/)).
- To improve the validity of this study, it is necessary to explain why different phenotypes are obtained between humans and mice by the deletion c.652del237bp.
The divergence between humans and mice has been explained in lines 98 to 117, after the mouse model evaluation. Various assumptions were made, and a more in-depth analysis has been added to fill this gap. In vitro experiments and in silico modeling are presented. A sentence has been added to the text.
Line 119-122: Taken as a whole, our findings highlighted the complexity of genetic regulation in fertility and underscored the need for further in vitro and/or in silico clarification of the role of the c.652del237bp deletion in human spermatogenesis. (line 113: using in vitro and/or in silico modeling)
- The results of this study only show the possibility and have not actually been examined on sperm. Are there any plans to conduct in vivo analysis using KO mice or to test using sperm from azoospermia patients?
We do not have access to the patient with c.652del237bp. Of the three patients in the literature, all have azoospermia and a negative testicular sperm extraction. Regarding the mouse model, homozygous females were obtained (see Ghieh et al, 2024), which suggests that the tex11 deletion does not prevent sperm production.
Minor points
- There are many “-” in the text. What does this mean (Line 87, 91, …)?
Done
- Fig. 4 is difficult to understand. How about explaining it in parts A, B, C, etc.?
Thank you for this comment. We have added a short paragraph to the Materials and Methods section and explain the results in more depth. We also differentiate between the different parts of the figures, so that the reader can better understand the data. Furthermore, the figure legend has been revised in accordance with your comment.
Material and methods section: Line 214-230
Results section: Line 257-269
Figures legends: Line: 274-288
- Fig. 4 should be included in the text, not in a subheading (3. In silico modeling) on page 7.
Done
Reviewer 2:
Thank you for your article. The pathogenic role of the recurrent TEX11 c.652del237 deletion in non-obstructive azoospermia (NOA) is indeed worth re-examining, and your combination of in vivo (mouse model), in vitro (HEK293 transfection), and in silico approaches makes this work particularly impressive. I have only a few minor recommendations.
Thank you for these positive comments.
- Introduction – The section on karyotyping and Y chromosome deletions could be shortened to allow readers to focus earlier on TEX11. At the same time, you may consider expanding the background on the TEX11 gene and protein to better highlight its biological and clinical relevance.
Thank you for the remark. In accordance, we have shortened the section on the karyotype and expanded the section on TEX11. Line 76-78, 92-102.
- Materials and Methods – It is excellent that you provide your protocol in such detail, which will facilitate reproducibility and verification of your work. However, some of this information may be too technical for the main text. You might consider moving the more detailed protocol to supplementary materials.
Thank you for this suggestion (which would be easy to do) but it goes against the comments from the editor, who requested “…adding full experimental details, presenting all the results completely”. I hope you will agree with the editor.
Discussion – You note that WES and WGS were not performed in some of the reported cases, which is an important limitation of earlier studies. Expanding on this point would strengthen the discussion and further emphasize the need for comprehensive genetic analysis.
As you suggest, we have expanded this section by adding a new sentence.
Line 330-334 : . Even though the presence of the in frame TEX11-c.652del237 variant might well explain the infertility of the three patients described in the literature, another genetic etiology would have to be ruled out with WES or WGS. Furthermore, aCGH is now an outdated diagnostic technique, given that WES and WGS can identify both copy number variations and single nucleotide changes.
Figures and Tables – No changes are needed; these are clear and appropriate.
Thank you
Reviewer 2 Report
Comments and Suggestions for Authors
Dear Authors,
Thank you for your article. The pathogenic role of the recurrent TEX11 c.652del237 deletion in non-obstructive azoospermia (NOA) is indeed worth re-examining, and your combination of in vivo (mouse model), in vitro (HEK293 transfection), and in silico approaches makes this work particularly impressive. I have only a few minor recommendations:
-
Introduction – The section on karyotyping and Y chromosome deletions could be shortened to allow readers to focus earlier on TEX11. At the same time, you may consider expanding the background on the TEX11 gene and protein to better highlight its biological and clinical relevance.
-
Materials and Methods – It is excellent that you provide your protocol in such detail, which will facilitate reproducibility and verification of your work. However, some of this information may be too technical for the main text. You might consider moving the more detailed protocol to supplementary materials.
-
Discussion – You note that WES and WGS were not performed in some of the reported cases, which is an important limitation of earlier studies. Expanding on this point would strengthen the discussion and further emphasize the need for comprehensive genetic analysis.
-
Figures and Tables – No changes are needed; these are clear and appropriate.
Author Response

(The authors gave the same response as above.)

Round 2
Reviewer 1 Report
Comments and Suggestions for Authors
I think that the revised manuscript has been improved.
Reviewer 2 Report
Comments and Suggestions for Authors
Dear authors, congratulations on your work, I have no further comments.